# Strong Southward and Northward Currents Observed in the Inner Plasma Sheet

YanYan Yang[1, 2], Chao Shen[3], and Yong Ji[4]

[1]Institute of Crustal Dynamics, China Earthquake Administration, Beijing, 100085, China

[2]Center for Satellite Application in Earthquake Science, China Earthquake Administration, Beijing, 100085, China

[3]Harbin Institute of Technology, Shenzhen, 518055, China

[4]Department of Mechanics and Engineering Science, Peking University, Beijing, 100871, China

*Correspondence to*: Chao Shen (Email: shenchao@hit.edu.cn)

**Abstract.** It is generally believed that field aligned currents (FACs) and the ring current (RC) are two dominant parts of the inner magnetosphere. However, using the Cluster spacecraft crossing of the pre-midnight inner plasma sheet in the latitude region between 10°N and 30°N, it is found that, during intense geomagnetic storms, in addition to FACs and the RC, there also exist strong southward and northward currents, which should not be FACs, because the magnetic field in these regions is mainly along the *XY* plane. Detailed investigation shows that both magnetic field lines (MFLs) and currents in these regions high dynamic. When the curvature of MFLs changes direction in the *XY* plane, the current also alternatively switches between southward and northward. To investigate the generation mechanism of the southward and northward current, we employed the analysis of energetic particle flux up to 1Mev. For energetic particle below 40kev, observations from Cluster CIS/CODIF (Composition and Distribution Function analyzer) is used. While for higher energetic particle, the flux is obtained by extrapolations of low energetic particle data through Kappa distribution. The result indicates that the most reasonable candidate for the origin of these southward and northward currents is the curvature drift of energetic particles.

## 1 Introduction

Abundant current systems existing in the Earth's magnetosphere play a very important role in energy transformation in different regions (Kuijpers et al., 2014). Recently, through simulations and observations, numerous studies have shown that the inner magnetosphere currents have a more complicated structure and distribution than originally thought. For example, in the low latitude, the magnetic field geometry can be altered significantly into tail-like during storm time (Tsyganenko et al. 2003); One or multi banana current can exist in the inner magnetosphere, which makes the link of the current systems more complicated (Liemohn et al. 2013). In the high latitude, field-aligned currents (FACs) have more sophisticated structures except the known large scale region 1 and region 2 currents (Mishin et al., 1997; Dunlop et al., 2015a; 2015b). Therefore, more work is still needed to reveal the true nature of these current systems.

The huge progress in satellite deployments makes it possible for direct observation of the inner magnetosphere current system. It is believed that the magnetosphere and ionosphere are linked through a ring current (RC) and FACs (e.g., Le et al. 2004; Zhang et al. 2011). Therefore, many investigations are mainly focused on these two current systems, from high (e.g., Iijima and Potemra 1976; 1978; Wang et al., 2006; Dunlop et al., 2015b) and low latitude, respectively (e.g., Vallat et al. 2005; Shen et al. 2014; Yang et al. 2016). The region from low to middle latitude, which is the key area for the inner magnetosphere current link, however, has received less attention. Cartoon plots and some statistical results (e.g., Le et al., 2004) show that FACs should be the dominant current in these areas. Through Cluster satellite observations, Vallat et al. (2005) pointed out that the RC could exist at middle (or even high) latitudes. Despite the results achieved by these various research efforts, so far, there are still no findings enabling a conclusion about the complete current morphology in low and middle latitudes. For example, are FACs and the RC the only currents in these regions? If there are other currents, what is the corresponding generation mechanism for them? To address these questions, the current distribution and magnetic field geometry during two storm events are investigated in the latitude regions from 10°N to 30°N.

In the following, we will use Cluster fluxgate magnetometer (FGM) (Balogh et al., 1997) data to conduct the analysis for two reasons: 1. the polar orbit of Cluster offers an opportunity to go through both the low-latitude and middle-latitude regions and 2. The configuration of the four Cluster satellites makes it possible to calculate the current via Maxwell-Ampère's law and obtain the magnetic field geometry. Moreover, in many previous works, it was thought that an asymmetric RC linked with the FACs, which is generally believed to occur during storm time, so storm events are our primary focus here.

Throughout this paper, solar magnetospheric (SM) coordinates are used. To better describe angles, spherical coordinates (θ, $\phi$) in the SM frame are also defined, i.e., the polar angle θ (0° $\leqslant$ θ $\leqslant$ 180°) is the angle between the $+ Z$ axis and the vector direction while the azimuthal angle $\phi$ (0° $\leqslant$ $\phi$ $\leqslant$ 360°) is anticlockwise rotated from the $+ X$ axis in the *XY* plane when seen from $+ Z$ axis. For current density analysis, the local cylindrical coordinate system ($j_\rho, j_\varphi, j_z$) (Vallat et al. 2005) is also utilized. Where $j_z$ is parallel to the +Z axis; $j_\rho$ represents the radial component of the current on the plane parallel to the X-Y plane, oriented anti-earthward; $j_\varphi$ points eastward, describing RC.

## 2 Methodology

In this study, magnetic curvature analysis (MCA) (Shen et al., 2003) and magnetic rotation analysis (MRA) (Shen et al., 2007) are used; these techniques have the unique ability to reveal the three-dimensional geometric structure of the magnetic

field directly as well as provide more detailed magnetic-field-related parameters, such as magnetic field gradient, curvature, and the binormal of magnetic field lines, rotation rates, and current density. The magnetic unit vector $\hat{\mathbf{b}} = \mathbf{B} / |\mathbf{B}|$, curvature vector $\vec{\rho}_c$ ($\vec{\rho}_c = (\hat{\mathbf{b}} \cdot \nabla)\hat{\mathbf{b}}$), and the binormal vector $\hat{\mathbf{N}}$ ($\hat{\mathbf{N}} = \hat{\mathbf{b}} \times \hat{\rho}_c / |\hat{\mathbf{b}} \times \hat{\rho}_c|$) are orthogonal to each other in the analysis, and the radius of curvature is $R_c = 1 / \rho_c$. The magnetic vector $\mathbf{b}$ has maximum, median, and minimum rotation rates of

$\mu_1^{1/2}$, $\mu_2^{1/2}$, and $\mu_3^{1/2}$ along $\hat{\mathbf{e}}^{(1)}$, $\hat{\mathbf{e}}^{(2)}$, and $\hat{\mathbf{e}}^{(3)}$, respectively, where $\hat{\mathbf{e}}^{(1)}$, $\hat{\mathbf{e}}^{(2)}$, and $\hat{\mathbf{e}}^{(3)}$ are the three characteristic eigenvectors of the magnetic field. Note that, because the strong geomagnetic field in the region of interest will produce artificial currents in the basic MRA calculation (nonlinear contributions), the dipole field is subtracted when using the MRA method to minimize truncation error (Shen et al., 2014).

To make a comparison with the nondisturbed geomagnetic field, the local dipolar values of magnetic field strength $B_{tDip}$,

radius of curvature, $R_{cDip}$, magnetic field gradient strength $|\nabla B_{Dip}|$, and three rotation rates $\mu_1^{1/2}$, $\mu_2^{1/2}$, and $\mu_3^{1/2}$ are also presented. They are calculated (Shen et al., 2014) by using:

$$B_{tDip} = Mr^{-3}\sqrt{(1 + 3\cos^2\theta)},$$

$$R_{cDip} = \frac{r}{3}\sqrt{(1 + 3\cos^2\theta)^3} / [|\sin\theta| \cdot (1 + \cos^2\theta)],$$

$$|\nabla B_{Dip}| = 3Mr^{-4} \cdot \sqrt{1 + \cos^2\theta(7 + 8\cos^2\theta)} / \sqrt{(1 + 3\cos^2\theta)} \ ,$$

( 1 )

$$\mu_1^{1/2} = \mu_\theta^{1/2} = 3(1 + \cos^2\theta) / [r(1 + 3\cos^2\theta)],$$

$$\mu_2^{1/2} = \mu_\phi^{1/2} = 3|\cos\theta| / [r\sqrt{(1 + 3\cos^2\theta)}],$$

$$\mu_3^{1/2} = \mu_r^{1/2} = 0,$$

where $M = m \cdot \mu_0 / 4\pi$ (with $m = 7.78 \times 10^{22}$ A$\cdot$m$^2$ being the earth's magnetic dipole moment) and $r$ is the radial distance in SM coordinates.

**3. Event Analysis**

The chosen events occurred, respectively, on 12 April 2001 and 31 March 2001. These were the two largest storms from 2001 to 2004 during which the four Cluster satellites had a small (best) tetrahedron separation distance (≤1000 km). The minimum Dst indexes for the two events were −271 and −387 nT, respectively. During the two events, Cluster was in the

pre-midnight sector and traversed the RC region vertically from the southern to northern hemispheres. The region of interested is in the northern hemisphere. Figure 1 gives the proton density and differential flux for $H^+$, $He^+$, $O^+$ during the concerned interval, which are obtained from Cluster Ion Spectrometer (CIS, Rème et al., 2001). The figure indicates that Cluster is mainly in plasma sheet region (e.g., Vallat et al. 2005).

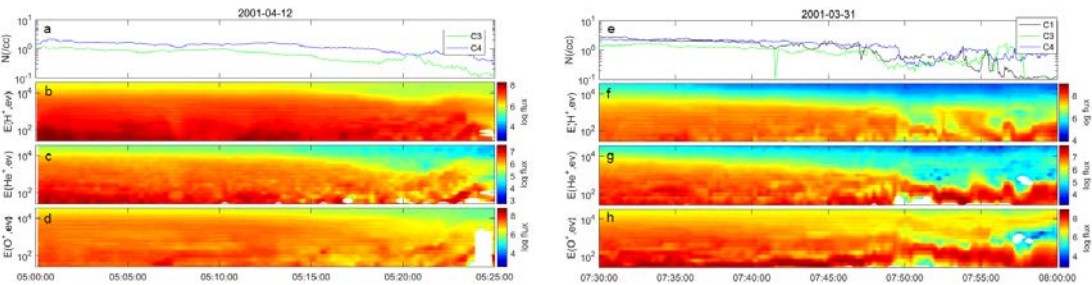

**Figure 1: Cluster CIS data for 12 April 2001 (left) and 31 March 2001 (right) event. (a, e) the proton density variation for three satellites C1 (black), C3 (green) and C4 (blue). (b-d, f-h) $H^+$, $He^+$, $O^+$ energy time spectrograms in particle flux units ions/(cm2 sr s Kev) from C4.**

### 3.1 12 April 2001 event

The time interval of interest for the first event is from 05:00 to 05:25 UT, with latitude ranging from 16.9° to 25.7°. Figure 2 presents some of the main physical quantities. Figure 2a shows the average magnetic field $\langle B_t \rangle$ detected from the four Cluster satellites and the local dipolar magnetic field strength. It can be seen that the local magnetic field is enhanced in this area. Figure 2b indicates that the polar angle of the magnetic field is close to 90°, indicating that the magnetic field lies approximately in the *XY* plane. The polar angle and azimuthal angle of dipolar fields is also show in dashed lines in Figure 2b, which indicates a large deviation of the polar angle with observations. Figure 2c shows that the radius of curvature, $R_c$, has large variations. It is interesting to see that $\phi_c$ (the angle of $R_c$ in Figure 2d) changes direction alternately during the whole period. Therefore, eight regions (numbered from NH1 to NH8) were chosen according to the changes in $\phi_c$ direction to investigate their features. The variations of some physical quantities are also summarized in Table 1. For $\phi_c$ and $\theta_{e1}$, the average values (with a few large abnormal points removed) during this period is given. '-' denotes that the value has large oscillations. For $j_z$, the maximum or minimum value during each interval is presented.

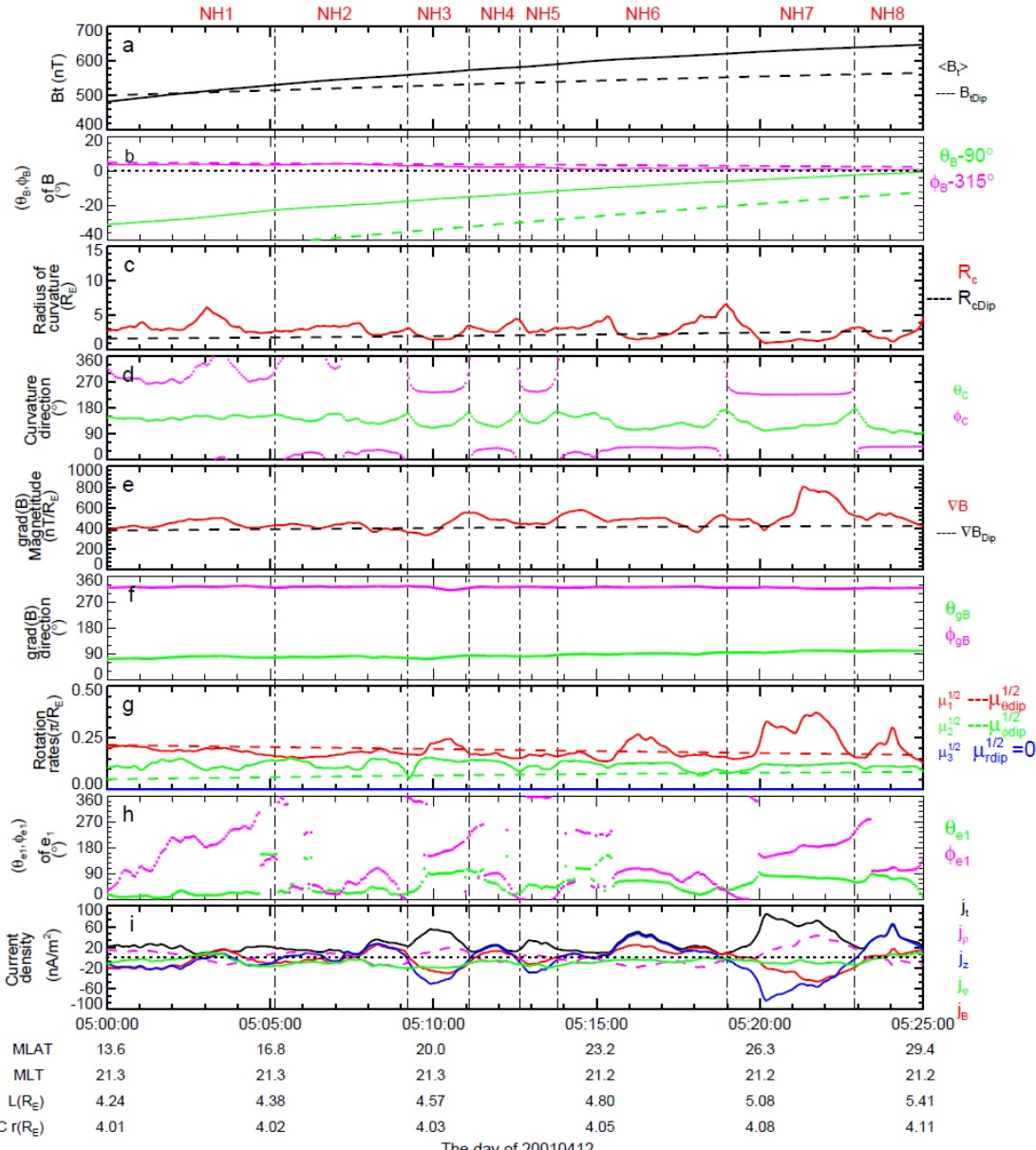

**Figure 2: Geometry of the magnetic field and the current distribution in the NH region on 12 April 2001. (a) Average magnetic strength** $B_t$ **at the center of the Cluster tetrahedron (black solid line) and the calculated strength** $B_{tDip}$ **of the dipole geomagnetic field (black dashed line). (b) Direction angles (** $\theta_B$ **,** $\phi_B$ **) of the magnetic field. 90°and 315º are reduced respectively for** $\theta_B$ **and** $\phi_B$ **to better indicate the magnetic field variation. The polar angle and azimuthal angle of dipolar fields is also show in dashed lines. (c) Radius of curvature,** $R_c$ **(red solid line), and the calculated radius of curvature,** $R_{cDip}$ **, of the dipole geomagnetic field (black dashed line). (d) Direction angles (** $\theta_c$ **,** $\phi_c$ **) of the curvature of the MFLs. (e) Value of the gradient of magnetic field strength for the real magnetic field (red solid line) and dipole geomagnetic field (black dashed line). (f) Direction angles (** $\theta_{gB}$ **,** $\phi_{gB}$ **) of the gradient of magnetic field strength. (g) Maximum, median, and minimum rotation rates of the measured magnetic field (solid lines)**

and dipole geomagnetic field (dashed lines). (h) Direction angles ($\theta_{e1}$, $\phi_{e1}$) of the maximum rotation rate. (i) Total current density $j_t$ (black line) and the three components $j_\rho$ (magenta line), $j_\varphi$ (green line), $j_z$ (blue line) in local cylindrical coordinate system, respectively. The red line is the field-aligned component $j_B$ (red line).

As shown in Figure 2c, the radius of curvature of MFLs in the eight regions is varied compared with that of the dipole field. Another feature observed in Figure 2c is that $R_c$ peaks at the vertical dashed lines. It is reasonable since the curvature radius in transition region should be larger than the region where the curvature radius has opposite directions. Figure 2d and $\phi_c$ row in Table 1 give the average value of the azimuthal direction $\phi_c$ during each interval. It quantitatively reveals that $\phi_c$ alternatively varied between 30.3°–51.9° and 230.3°–292.0°. It is noted from Figure 2d that, for some regions, the

variation of polar angle $\theta_c$ has larger fluctuation (than azimuthal angle $\phi_c$). This feature reflects larger changes of the magnetic field in Z component. Figure 2g shows that $\mu_1^{1/2}$ has an enhancement in each region, illustrating a stretched MFL structure. Figure 2h and row $\theta_{e1}$ in Table 1 show that, for most regions, the largest value of the polar angle $\theta_{e1}$ for $\mu_1^{1/2}$ is close to 90°; therefore, the largest deviation of MFLs is along the *XY* plane. Figure 2i indicates that the current oscillates and that the dominant current is along $j_\rho$ and north (or south) $j_z$ direction, while $j_\varphi$ is basically small compared with

$j_\rho$ and $j_z$. To show FACs, $j_B$ component is also given in Figure 2i, it can be seen that the value of $j_B$ close to that of $j_\rho$, because the direction of the magnetic field points approximately to the radial direction (see Figure 2b). The maximum values for $j_B$ and $j_z$ were ~40 and ~80 nA/m$^2$, respectively. From Table 1 and Figure 2, it is interesting to see that, from region NH1 to region NH8, the $j_z$ component changed from positive (northward) to negative (southward) as $\phi_c$ varied from <50° to >230°.


**Table 1: Variation of physical quantities for two storm events**

| Event[a] | PQ[b] | NH1[c] | NH2[d] | NH3[e] | NH4[f] | NH5[g] | NH6[h] | NH7[i] | NH8[j] | NH9[k] | NH10[l] | NH11[m] |
|---|---|---|---|---|---|---|---|---|---|---|---|---|
| | $\phi_c$ (°) | 292.0 | 41.4 | 244.1 | 35.3 | 251.9 | 36.9 | 230.3 | 44.8 | | | |
| 20010412 | $\theta_{e1}$ (°) | 29.5 | 27.0 | 74.7 | 57.7 | 51.9 | 61.0 | 70.8 | 69.7 | | | |
| | $j_{zm}$ (nA/m$^2$) | -22.5 | 27.2 | -50.8 | 23.3 | -28.8 | 46.6 | -82.6 | 63.1 | | | |

| | | | | | | | | | | | | |
|---|---|---|---|---|---|---|---|---|---|---|---|---|
| | $\phi_c$ (°) | 59.9 | 241.9 | 59.6 | 244.7 | 58.5 | 240.3 | 63.2 | 235.1 | 60.5 | 238.6 | 62.8 |
| 20010331 | $\theta_{e1}$ (°) | 71.3 | - | 65.4 | 73.2 | 71.8 | 59.8 | 73.4 | 71.7 | 78.8 | 59.9 | 80.2 |
| | $j_{zm}$ (nA/m$^2$) | 106.9 | −42.5 | 60.1 | −128.3 | 95.9 | −126.3 | 198.2 | −294.3 | 118.2 | −193.9 | 204.7 |

[a]Storm events considered in this work.

[b]The physical quantity $\phi_c$ is the average azimuthal direction of the curvature radius, $\theta_{e1}$ is the average polar angle of maximum rotation rates of the magnetic field, and $j_{zm}$ represents the maximum or minimum value of the $j_z$ current component.

[c–m]Regions for each storm event.

### 3.2. 31 March 2001 event

Another larger storm occurred between 07:30 and 08:00 UT on 31 March 2001. The event was once reported by Shen et al. (2014), but they only concentrated on the interval from ~07:00 to 7:25 UT. Observations are shown in Figure 3 for the latitude region from 13.1°N to 31.2°N, the interval during the main phase of the storm. Here, 11 regions designated from NH1 to NH11 are divided also according to azimuthal direction changes of $\phi_c$. The variations of some relative physical quantities are also shown in Table 1. From Figure 3 and Table 1, it can be seen that these parameters behave as same as that of the first event, but with strong magnetic field strength. Figure 3 indicates that the magnetic field strength is stronger than that during the first event. The magnetic field is in the *XY* plane (see Figure 3b). The radius of curvature of MFLs (see Figure 3c), the magnetic field gradient (Figure 3e). And the largest rotation rate (Figure 3g) oscillates significantly and exhibits large deviations compared with those of the dipole field. Figure 3f shows that the magnetic field gradient is in the *XY* plane and directed toward the dayside. Figure 3h and row $\theta_{e1}$ demonstrate that the largest variation of MFLs is near the *XY* plane. In Figure 3i, it is clear that the $j_z$ component is the dominant current, with a maximum value of ~300 nA/m$^2$. This value is more than triple that of the 12 April 2001 event. It is clear to see that the $j_\varphi$ component is the smallest among these currents. Similar to first event, $j_z$ is simultaneously observed to vary from northward to southward when $\varphi_c$ changes direction.

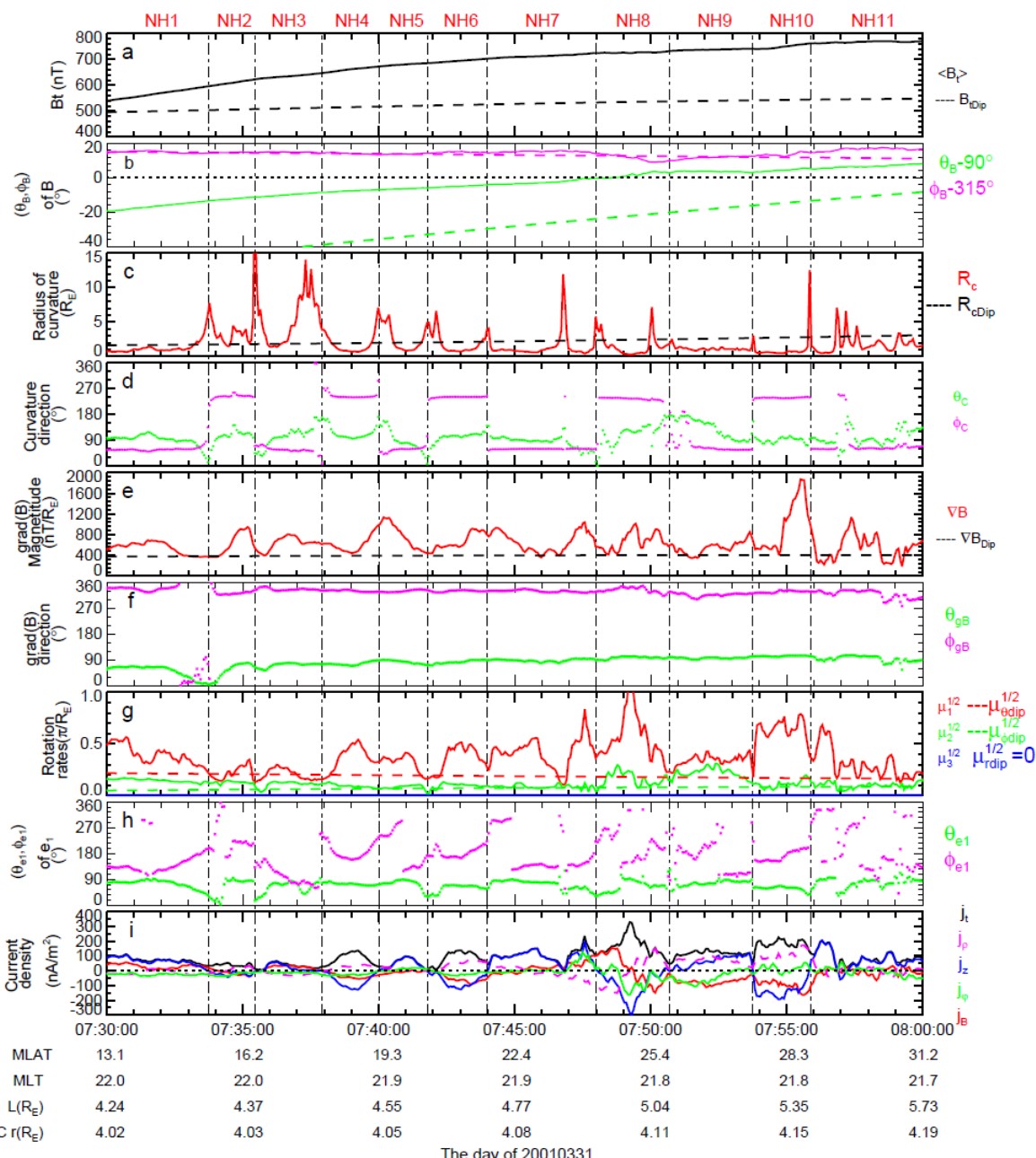

**Figure 3: Geometry of the magnetic field and the current distribution in the NH region on 31 March 2001. The format is the same as that of Figure 2.**

**4 Discussion**

5    During the 12 April 2001 and 31 March 2001 strong storm events, the Cluster satellites were located in the pre-midnight sector and crossed from ~10°N to ~30°N. In these regions, both the magnetic field parameters and the current density fluctuated significantly. The MFLs, which were mainly in the *XY* plane, severely deviated from the dipole field and changed (stretched) along the *XY* plane. Figure 4 displays the total magnetic field strength and its three components. It can be seen that the *X* and *Y* components of the magnetic field have the largest fluctuations, which is consistent with the results obtained

from Figure 2 and 3. To further investigate the fluctuation, the continuous 1-D wavelet transform method is applied in X and Y component of the magnetic field. It is found that the ULF wave covering a range of frequencies spanning 4 mHz to 10 mHz can be observed (not shown here), which is consistent with the typical current density variation in ~2-4min period. Actually, ULF wave in the plasma sheet region has been extensively reported in previous works (see Keiling, 2009 and references therein). So, it seems that ULF wave is a possible way to cause the variation of curvature radius (and the field aligned current).

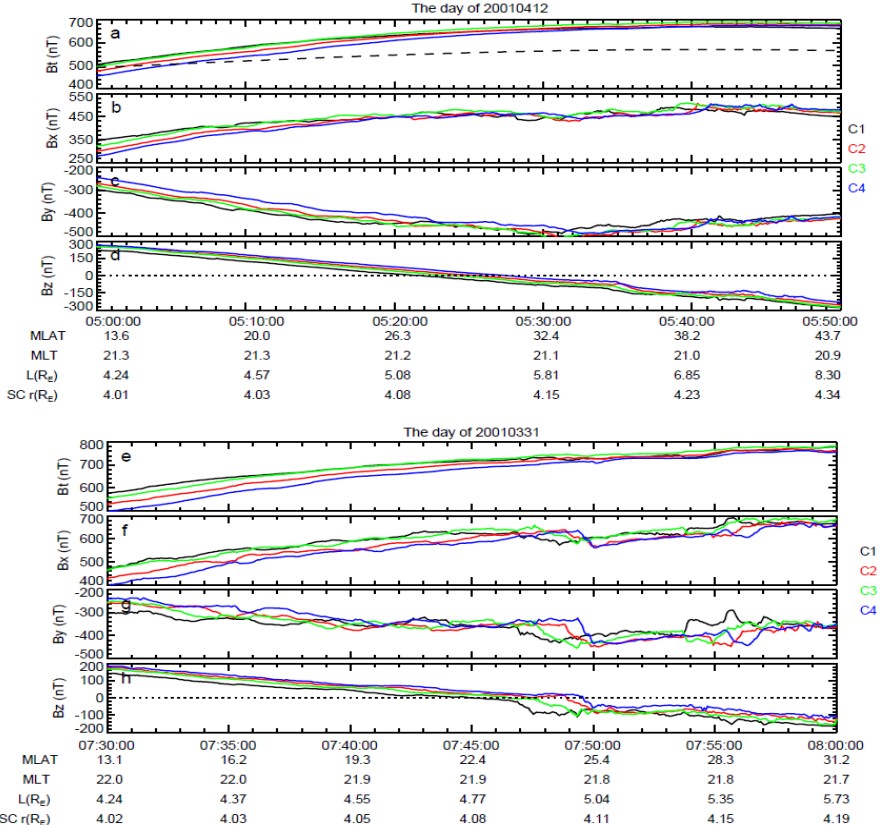

**Figure 4: Magnetic field observed by the four Cluster spacecraft during 12 April 2001 and 31 March 2001 storm events.**

The most obvious phenomenon in the two cases is that the existence of three current systems, i.e., FACs $j_B$, an azimuthal current $j_\varphi$ and a northward (or southward) current $j_z$. Among them, $j_z$ is basically the strongest current component. In previous studies (e.g., Le et al., 2004; Vallat et al., 2005), the existence of $j_B$ and $j_\varphi$ has been proved. However, the occurrence of such a strong $j_z$ in the inner plasma sheet has not been reported before. In the work of Vallat et al., (2005), they also found a southward current (see Fig 14 and corresponding text). But it is in equatorial ring current region (with no direction changes) and mainly caused by an asymmetry between the ionospheric conductivities of the two hemispheres. It is

very clear that the southward current in their paper is different with what we report here.

As introduced in previous studies (e.g., Parker, 1957), the current in the inner magnetosphere generally arises from gradient drifts as well as curvature drift and the gyromotion of energetic particles. They can be calculated by using (e.g., Lui et al., 1987; De Michelis et al., 1999):

$$\mathbf{j}_\nabla = P_\perp \frac{\mathbf{B} \times \nabla B}{B^3},$$
( 2 )

$$\mathbf{j}_C = -\frac{P_\parallel}{B^2} \boldsymbol{\rho}_c \times \mathbf{B},$$
( 3 )

$$\mathbf{j}_G = \frac{\mathbf{B}}{B^2} \times \left[ \nabla P_\perp - \frac{P_\perp}{B} \nabla B - \frac{P_\perp}{B^2} (\mathbf{B} \cdot \nabla) \mathbf{B} \right],$$
( 4 )

where $\mathbf{j}_\nabla$, $\mathbf{j}_C$, and $\mathbf{j}_G$ represent the gradient current, curvature current, and gyromotion current, respectively, and $P_\perp$ $P_\parallel$ are the pressure tensor components perpendicular and parallel to the magnetic field, which can be deduced from:

$$P_\perp = \pi\sqrt{2m} \iint J\sqrt{\varepsilon} \sin^3 \alpha \, d\alpha \, d\varepsilon$$
,
( 5 )

$$P_\parallel = 2\pi\sqrt{2m} \iint J\sqrt{\varepsilon} \cos^2 \alpha \sin \alpha \, d\alpha \, d\varepsilon$$
,
( 6 )

where m is mass of particle, $J$ is the differential flux intensity, $\varepsilon$ and $\alpha$ are the particle energy and pitch angle, respectively. Since the magnetic field gradient $\nabla B$ and curvature $\boldsymbol{\rho}_c$ have been obtained by using MRA method, the above three currents can be calculated when the pressure tensor components are given.

15    For the two events in this study, both the magnetic field and magnetic field gradient are directed toward the dayside. Therefore, the current deduced from $\mathbf{B} \times \nabla B$ (the gradient drift current) should be small. To analyze the current contribution from gyromotion drift and curvature drift, we first show the three components of $-\boldsymbol{\rho}_c \times \mathbf{B}$ for the two events in Figure 5a and 5b. It is clearly seen that the $(-\boldsymbol{\rho}_c \times \mathbf{B})_z$ component is the dominate part and has the same variation trend with $j_z$. Therefore, the curvature drift current is a possible candidate. For gyromotion current, it is originated from three

20    terms, i.e., $\mathbf{B} \times \nabla P_\perp$, $-\mathbf{B} \times \nabla B$ and $-\mathbf{B} \times (\mathbf{B} \cdot \nabla) \mathbf{B}$. Firstly, according to previous works (e.g., Lui et al., 1987; De Michelis et al., 1999), $\nabla P_\perp$ is along the radial direction, which means that it has the similar direction with magnetic field for two

events concerned here. So, the contribution from $\mathbf{B}\times\nabla P_\perp$ should be small. Secondly, $-\mathbf{B}\times\nabla B$ is similar to the gradient

drift current and can be negligible. Thirdly, since $(\mathbf{B}\cdot\nabla)\mathbf{B}$ has the same direction with $\boldsymbol{\rho}_c$ ($\boldsymbol{\rho}_c=(\hat{\mathbf{b}}\cdot\nabla)\hat{\mathbf{b}},\hat{\mathbf{b}}=\mathbf{B}/|\mathbf{B}|$),

according to Figure 5a and 5b, the product of $-\mathbf{B}\times(\mathbf{B}\cdot\nabla)\mathbf{B}$ (similar to $\boldsymbol{\rho}_c\times\mathbf{B}$) will behave oppositely to $j_z$.

Consequently, the gyromotion current has little possibility of contributing to a strong $j_z$. According to the above analysis,

5    the most reasonable candidate for strong $j_z$ should be the curvature drift.

    Based on the above analysis, cartoon plots are given in Figures 5c and 5d to explain the possible generation mechanism for

$j_z$. During the strong storm time, turbulences, e.g., ULF waves, result in the fluctuation of the MFLs, then, the radius of

curvature of the MFLs decreases, leading to an increase in the curvature drift current. During this process, the direction of

the magnetic field is nearly unchanged because the background field is very strong. However, the curvature will alternately

10    change directions along with the variation of the MFLs, resulting in alternating variations of $-\boldsymbol{\rho}_c\times\mathbf{B}$, i.e., leading to the

oscillation of $j_z$.

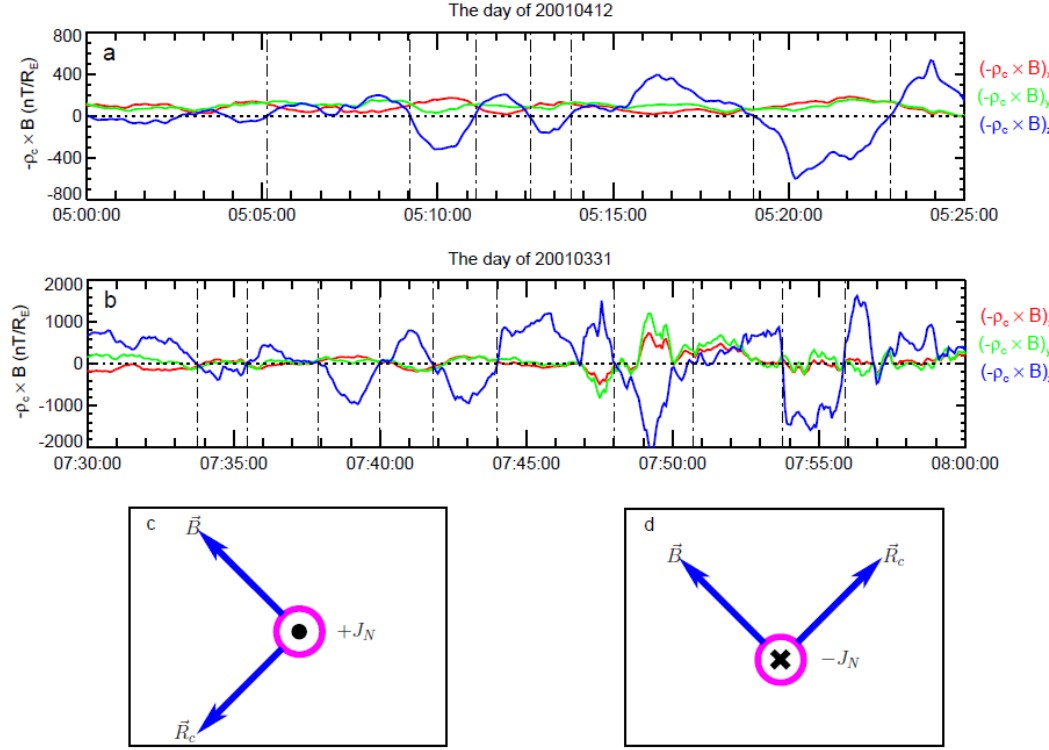

**Figure 5: (a) and (b) The three components of $-\boldsymbol{\rho}_c\times\mathbf{B}$ for two concerned events. $\boldsymbol{\rho}_c$ is calculated from MRA method and $\mathbf{B}$ is**

**the averaged magnetic field measured by four Cluster. . (c) and (d) Cartoon plots of the origin of the $j_N$ current variation.**

Figure 5 (a) and (b) can only illustrates that the direction of $-\rho_c \times \mathbf{B}$ is consistent with northward current. To quantitatively check if curvature current calculated through Eq. (3) is consistent with result obtained from MRA method, further investigation deserve to be tried. The CIS/CODIF (Composition and Distribution Function analyzer) can provide the differential flux intensity for energy below 40kev. Through Eq. (3) and (5, 6), the curvature current can be estimated. The result shows that the main variation trend is consistent with result from MRA, but the intensity is very small (less than 1 nA/m$^2$, not show here). However, it should be noted that, for Cluster CIS/CODIF, only low energetic particle data are available, therefore, large bias may exist when calculating storm time current. In contrast, much higher energy is used in previous studies (e.g., to 1Mev in the work of Lui et al., 1987). Cluster RAPID can provide energy spectrograms for high energetic particle from ~27.6kev to ~3056 kev. Unfortunately, there is no available data for the two concerned events. The statistical study from Kronberg et al. (2015) proves that, in the near earth plasma sheet, higher energetic hydrogen and oxygen are greatly enhanced during geomagnetic activity. In the work of Ma et al. (2012), they also indicated that the flux for higher energetic particles could comparable or larger than that of the low energetic particles.

Though, there is no available differential flux for high energetic particles on Cluster, the curvature current still can be estimated through simulations. Previous works has proved that the particle distribution in plasma sheet can be described as Kappa distribution functions (Pierrard and Lazar, 2010, and references therein):

$$f = N_1 \left( \frac{1}{2\pi m E_0 \kappa_1} \right)^{2/3} \frac{\Gamma(\kappa_1 + 1)}{\Gamma(\kappa_1 - 1/2)} \left( 1 + \frac{E}{\kappa_1 E_0} \right)^{-\kappa_1 - 1} \tag{7}$$

Where $N_1$ and $E_0$ denotes to particle density and temperature, and $\kappa_1$ is a constant. For energy satisfying $E \gg E_0$, Eq. (7) can be written as:

$$f = aE^{-\kappa_1 - 1} \tag{8}$$

Since the differential flux intensity $J$ and particle velocity distribution function $f$ is related by $J = fp^2$, Eq. (8) is also the function of $J$, namely:

$$J = ap^2 E^{-\kappa_1 - 1} \tag{9}$$

Where $p$ is the momentum of the concerned particles, and $a$ is a constant. Thus, with the known differential flux intensity from low energetic particle, the parameter $a$ and $\kappa_1$ can be determined. Then, the differential flux intensity for high energetic particles (to 1Mev) can be estimated using Eq. (9). Though, particles are accelerated during the storm, we

have confirmed that the Kappa distribution is still satisfied using CIS/CODIF observations (not shown here). However, it should be noted that, during the storm, $a$ and $\kappa_1$ are no longer a constant but varied with time. Besides, to check if the estimated high energetic particle differential flux (using low energy particle data) is reasonable, we select a storm event occurred on 20 April 2002, which has similar position with two concerned events in this study, and has CIS/CODIF and

RAPID observations at the same time. The result shows that the fitted result (from CIS/CODIF measurement) can basically reflect the main trend of the high energetic particles, which can demonstrate that our estimation used here is reasonable. During the storm time, currents calculated via energetic particle fluxes appear to still underestimate the current. As the particle flux fit method of calculating currents works so well earlier in the time period, this undershoot during storm time might be indicative of additional energetic particle acceleration (a harder power law) in the parallel direction. This

increased parallel pressure would result in the observed larger value of $j_c$.

Now, we can re-estimate the curvature current using Cluster CIS/CODIF observations for energy between 25ev-40kev and simulation values for energy >40kev-1Mev. Figure 6 shows the estimated z component of curvature current (the red dotted curve). It is close to result from MRA (the blue curve).

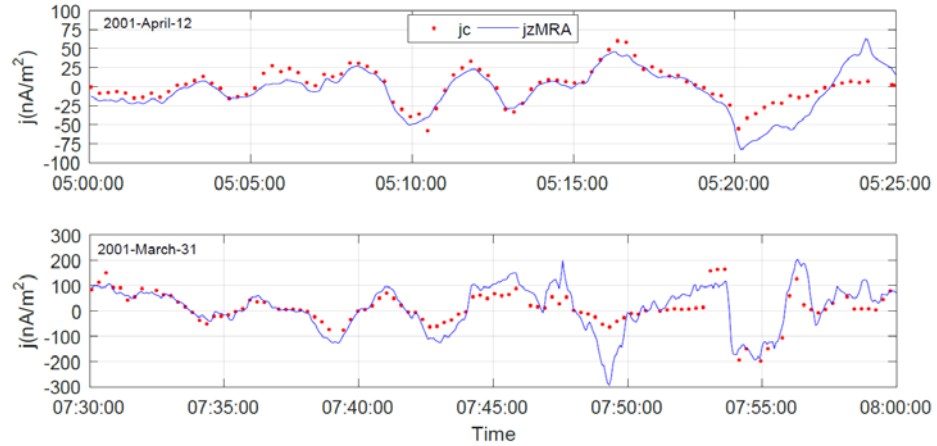

**Figure 6: The z component current density calculated from MRA method (the blue curve) and the estimated curvature current (the red dotted curve).**

It should be noted that both two events concerned here is in the north hemisphere. Actually, we have checked that the southward and northward current also can be observed in the southern low and middle latitudes. So, such currents should be observable both in north and south inner plasma sheet during strong geomagnetic storm events.

According to previous analysis from plasma data (Baker et al., 2002; Korth et al., 2004; Vallat et al., 2005; Ohtani et al., 2007), most NH regions should correspond to the plasma sheet region. Using the T96 model (Tsyganenko, 1995, 1996), we have tried to trace Cluster footprints in the northern hemisphere, it is found that the position is ~55°–60°(not shown here),

which just corresponds to the position of the FACs (Papitashvili et al, 2002; He et al, 2012). Because the MFL shapes in the plasma sheet have been changed considerably, the particle motion in Earth's magnetic field will be altered correspondingly, which may affect the particle distribution in the polar and equatorial regions, hence, leading to the variation of the FAC and RC distributions. These effects, however, need to be evaluated in future work.

5    When calculate current density using MRA method, it should be noted that Cluster is not a regular tetrahedron shape around the perigee area, but suffers to an elongation, which can produce an unnatural currents. These unnatural currents are included in our analysis and cannot be removed. To evaluate this component, methods from Robert et al. (1998) and Vallat et al. (2005) are used. Figure 7 gives the Cluster tetrahedron parameters for two concerned events. Then, the current influence of the tetrahedron shape can be estimated as a function of elongation and planarity (Figure 7c and Figure 7d). It can be seen that the

10   error caused by tetrahedron is never more than 30%.

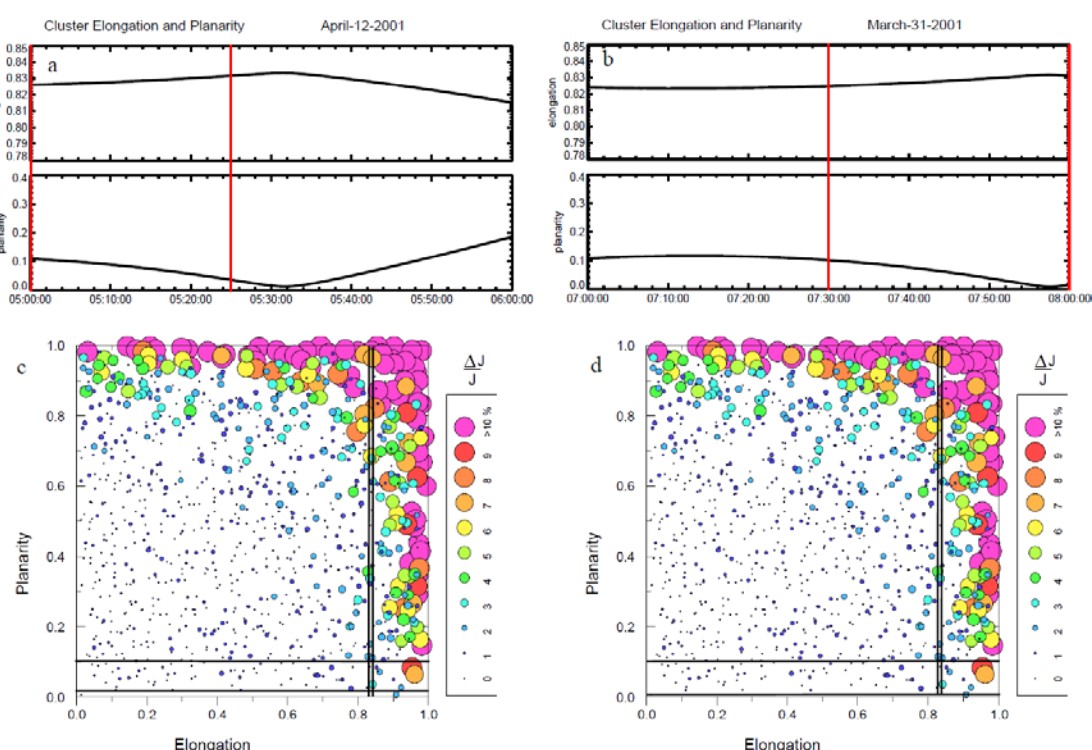

**Figure 7: Panel a and b: Cluster elongation and planarity for two studied events. The red vertical lines demarcate the concerned time interval. Panel c and d are picked from Robert et al. (1998) to evaluate the influence of the tetrahedron shape. Black lines**

15   **mark the elongation and planarity obtained from panel a and b.**

**5 Sumamry**

In this work, the magnetic field geometry and current density in the inner plasma sheet during two intense geomagnetic storms have been investigated. It is found that, the magnetic field and current density are highly fluctuated in this region.

Generally, both three components of current can be observed during the concerned interval. However, the northward (or southward) current is basically the strongest one. Detailed study shows that, the MFLs line in XY plane, so, the northward (or southward) current should not be FACs. This property has not been reported before.

The most prominent feature of the northward (or southward) current is the alternative changing of its direction, which is found to vary simultaneously with that of the curvature. To reveal the generation mechanism of the northward (or southward) current, gradient current, curvature current, and gyromotion current are analyzed, respectively. The result shows that the curvature current has the same variation trend with the northward and southward current. Then, using low energetic particle observations from Cluster CIS/CODIF, combined with simulations based on Kappa distribution, the curvature current is calculated. It shows that the estimated curvature current coincides very well with the current density directly obtained from MCA and MRA. Therefore, the curvature drift of the energetic particle is the most reasonable mechanism of the southward and northward current.

For the two events concerned in this work, we can observe ULF waves, which is consistent with the typical current density variation period. These turbulences excited during the strong storm can result to the decrease of curvature radius and changing of direction of MFLs, then leading to an increase of the curvature currents and variation of their direction.

**Acknowledgments and Data**

This work was supported by National Key R&D Program of China (Grant no. 2018YFC1503501) and the National Natural Science Foundation of China (Grant Nos. 41874190, 41231066 and 41204117). Solar wind data (OMNI data set) were obtained from http://omniweb.gsfc.nasa.gov; Cluster FGM and CIS/CODIF data were obtained respectively from ftp://cdaweb.gsfc.nasa.gov/pub/data/Cluster/, https://www.cosmos.esa.int/web/csa.

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
