# Peer review of "Strong Southward and Northward Currents Observed in the Inner Plasma Sheet"

_Annales Geophysicae, 2019_

## Referee Comment (RC1) · Anonymous Referee #1 · 13 Jun 2019

Reviewer's report on angeo-2019-56 Strong Southward and Northward Currents Observed in the Inner Plasma sheet By Yanyan Yang et al. This is an interesting paper for investigating the currents at low and middle latitudes during intense geomagenetic storms. The results showed that there exist also southward/northward currents in the inner plasma sheet, but they are neither the ring currents nor field aligned currents. The authors suggest that such horizontal currents at low and middle latitudes are caused by the curvature drift of energetic particles during magnetic storms. In general, the paper is well written, and the observational results support their conclusion. However, before it is accepted by Annales Geophysicae, some comments listed below may need to be taken into account.

Major comment 1. The authors focus only the northern middle latitudes in their study.

[Figure]

One interesting question would be to check the southern low and middle latitudes, to see if similar currents can also be observed. If yes, it might indicate that such currents are field-aligned. If not, the authors could provide some explanation or suggestions. 2. It seems there is no real summary in the end of the study. The author may think to add a typical summary.

Some minor comments Abstract: 1. during large storm events -> during intense geomagnetic storms 2. which cannot be FACs -> which should not be FACs 3. highly fluctuate -> high dynamic

Introduction 4. Page 1, line 20: "Recently, through simulations and observations, numerous studies have shown that the inner magnetosphere currents have a more complicated structure and distribution than originally thought". The author are suggested to provide more detailed description of "more complicated structure and distribution than originally though", not just added references there. 5. Page 1, line 26: respectively, from high latitude and low latitude -> from high and low latitudes, respectively. 6. Page 2, line 6: in the latitude regions from 10°N to 50°N In the abstract, the authors claimed that they focus on the latitude range between 10-30° N.

3. Event analysis 7. Page 7, line 7: (The event was once reported by Shen et al. (2014), but they only concentrated on the interval from ∼07:00 to 7:25 UT). -> The event was once reported by Shen et al. (2014), but they only concentrated on the interval from ∼07:00 to 7:25 UT. 8. Page 7, lines 11-12: It can be seen that these parameters behave as same as that of the first event, but with stronger magnetic field strength. 9. page 7, line 14: And the largest rotation rate (Figure 3g) oscillates significantly and exhibits. . .

4. Summary and Discussion It should be Discussion and Summary 10. Page 10, line 12: $\varepsilon$ and $\alpha$ are the particle energy and pitch angle, respectively. 11. Page 11, line 11: During the strong storm time, turbulences, e.g., the ULF waves, result in the fluctuation of the MFLs, . . . 12.Provide a typical summary of this work at the end of the study.

Please also note the supplement to this comment:
https://www.ann-geophys-discuss.net/angeo-2019-56/angeo-2019-56-RC1-supplement.pdf

---

## Referee Comment (RC2) · Anonymous Referee #2 · 19 Jun 2019

The authors present an original research manuscript which aims to explain observations performed in the northern nightside magnetosphere, where northward and southward currents are detected along with oscillations in the field curvature. In this region, the dominant currents are field-aligned currents and the ring current, which should exist mostly in the X-Y-plane. The authors propose that these currents are due to curvature drifts of energetic particles. The authors base their analysis on two mid-latitude storm time vents and Cluster spacecraft data, and various analysis techniques using multiple instruments from the Cluster mission.

The manuscript has generally good structure, a well-chosen selection of figures and good language. The authors take into account some of the important potential sources of errors in their analysis. The topic of research is interesting, and indeed worth pur-

suing. However, I have some fundamental questions regarding the logic behind this analysis and the application of the methods, and would appreciate if the authors could substantially clarify the issue, as well as improve the methods which would make the results more robust.

Major issue: The authors present MRA-method analysis of magnetic curvature and compare it with measured current densities, postulating that the current is due to curvature drift. However, the whole method of measuring currents aboard Cluster is based on the Curlometer technique, which calculates the curl (curvature) of a magnetic field and calculates the current from that. I feel that the authors need to give more reasoning as to why they consider it a new finding that one magnetic curvature analysis technique explains currents calculated via another magnetic curvature analysis technique.

Major issue: page 6 lines 13-17: If I'm interpreting this correctly, the coordinate system in use ($j_B$, $j_N$, $j_R$) changes constantly with the magnetic field measurements (as referred to as a local natural coordinate system in the caption of Figure 2). Is this calculated pre- or post IGRF (or dipole field?) deduction? I'm worried that the coordinate system is not well defined when the direction of the curvature changes abruptly. Looking at the evolution of the theta and phi angles for the direction of curvature, it looks to me like $j_N$ is mostly in the east-west-direction, and has a north-south component only at the extrema of the curvature oscillation. A re-decomposition of currents into cartesian or SM coordinates would have been important to answer questions arising from this coordinate selection, and would help with evaluation of results.

A question that arises directly from this, is how much of the observed $j_N$ is actually new current density in the north-south direction, and how much of it is existing ring current simply re-mapped due to an abruptly changing coordinate system?

Major issue: page 12, description of how the authors evaluate plasma pressure via the kappa distribution and CIS/CODIF: I believe the authors should further clarify how they perform this analysis in order to quell concerns regarding the trustworthiness of the

method. I shall try to elaborate. A Kappa distribution behaves wholly differently (quasi-Maxwellian) at low energies where CODIF data is available, than at higher energies, in the tail. As seen in Figure 1, the energy spectrogram varies wildly during the storm, and thus, assuming the plasma to be in something like a steady state and describable with a kappa distribution is a bold suggestion. After all, during storms is when there is strong acceleration of particles and deviation from the mean distribution.

The authors could explain in better detail how is the fit performed exactly, and what are the deduced kappa values? Both parameters "a" and "kappa" are being fitted using the low-energy portion of the population - some estimation of the quality and reliability of this fit should be presented.

Also, is the population assumed to be isotropic? Equation (3) takes only the parallel pressure, after all. Figure 6 shows the final result, but comparing that with Figure 5 suggests that the pressure contribution to the calculated current is minimal, and rather, it is dominated by the curvature component (as shown in Figure 5). And if the result is dominated by curvature, then of course it will match up with the MRA method (and the inherent curlometer technique). Thus, I am not convinced that energetic particle curvature drifts are particularly important here.

Minor issues / clarification requests:

page 2 line 9: Although others have called the curlometer technique "direct" measurement of the current, in truth, direct calculation would be counting charged particle fluxes. Perhaps briefly state that it uses magnetic curvature to calculate currents via Maxwell-Ampère's law.

page 2 line 20: A normal can be defined for a plane, not directly for a field line, unless you assume it to be in a plane within which the local curvature is. Please clarify.

page 2 lines 24-26: Could the authors please clarify, why they state that they subtract the IGRF field, yet then proceed to describe the standard dipole formulation?

page 5, Figure 2: Could the authors please explain why they plot a dipole field for comparison, instead of the IGRF field they state they use in the text?

page 6 line 5: Evaluation of Figure 2 panel c shows that contrary to what is written here, the radius of curvature is nearly everywhere much greater than that of the dipole field. Only in NH3 and NH6-8 does it drop below the dipole field value, and only then in the middle of the domain.

page 8 line 7: Stating that the field lines were severely deviated would be more readily confirmed had the authors included x,y,z components of the magnetic field. The radius of curvature is a challenging method of showing this, as it becomes most important at very small values, which are not clearly visible in the plots.

page 10 lines 16-18: The text should reference Figure 5, panels a and b. I would recommend stating more clearly what is being shown and analyzed, instead of simply referring to "a result", which here is simply the cross product of the curvature and the magnetic field. Also, the authors claim that the z component of this has the same variation trend as $j_z$, but $j_z$ has not been shown in any figure. If the authors claim that this is the same as $j_N$, the questions regarding stability of the chosen coordinate system apply again. I think the manuscript would be much improved if these doubts could be clarified.

page 10 line 19-21: I believe the authors should clarify their reasoning for disregarding the possibility of the third term of gyromotion drift to cause currents in the $j_z$ direction. On line 15, they stated that both the magnetic field and its gradient are pointed towards the dayside, so this term might be non-negligible.

Figure 5: The caption states that the plot shows "results deduced from the radius of curvature of the cross magnetic field" - I would recommend the authors be more explicit and exact in their statements.

page 13, line 6: Now the authors compare with the T96 model, but provide no refer-

ence. Does comparison with the T96 model provide some benefit over using the IGRF or dipole models, which are used(?) in the rest of the analysis? Remaining consistent would improve the readability of the manuscript.

page 13, lines 12-16: The error caused by planarity or elongation of the tetrahedron could do with a clear statement that deformation remains low. If I have understood correctly, neither the standard curlometer technique nor the MRA method attempt to remove the error, and this could be clarified.

page 14, Figure 7: The caption should be improved - what are the red vertical lines in panels a and b? Apparently the cross-lines in panels c and d indicate the region applicable for these two events, but this could be clearly stated - it looks like the panels were identical at first glance.

Technical corrections:

page 10 line 4: The reference is incorrectly formatted; it should read "De Michelis et al., 1999"

---

## Author Comment (AC2) · 22 Jul 2019

Since our response includes many equations and too many plots, we will include response (along with change-noted manuscript and revised manuscript) in supplement.

Please also note the supplement to this comment:
https://www.ann-geophys-discuss.net/angeo-2019-56/angeo-2019-56-AC2-supplement.zip

---

## Author Response (AR1)

**Response to Referee #1**

We appreciate all comments from the reviewer, which help to further improve the quality of our manuscript. In this round of revision, we have considered all comments seriously. The point-by-point revisions have been made and tracked in the change-noted manuscript. We hope that the new version of manuscript has met the requirements from the referee and ANGEO. In the following, the comments are marked in bold Times New Roman, followed by our responses.

**This is an interesting paper for investigating the currents at low and middle latitudes during intense geomagenetic storms. The results showed that there exist also southward/northward currents in the inner plasma sheet, but they are neither the ring currents nor field aligned currents. The authors suggest that such horizontal currents at low and middle latitudes are caused by the curvature drift of energetic particles during magnetic storms.**
**In general, the paper is well written, and the observational results support their conclusion. However, before it is accepted by Annales Geophysicae, some comments listed below may need to be taken into account.**

**Major comments:**

**1. The authors focus only the northern middle latitudes in their study. One interesting question would be to check the southern low and middle latitudes, to see if similar currents can also be observed. If yes, it might indicate that such currents are field-aligned. If not, the authors could provide some explanation or suggestions.**

Response:

Thank you for your good suggestion. Actually, the southward and northward current also can be observed in the southern low and middle latitudes. The following two figures show the geometry of the magnetic field and the current distribution in the southern hemisphere for the two events concerned in this work. We can see similar

fluctuations with what we observed in the manuscript.

So, the reported southward and northward current can observed both in north and south hemisphere. However, they are not the field-aligned currents. Actually, we have also given field-aligned current component ($j_B$) in last panel of the figure. The observed southward and northward current is the component perpendicular to $j_B$.

Anyway, it is a good suggestion to mention result from south hemisphere. We have added a short discussion in 'Discussion' part, see lines 14-16 in page 16 of the change-noted manuscript.

[Figure]

[Figure]

**2. It seems there is no real summary in the end of the study. The author may think to add a typical summary.**

Response:

Thank you for your suggestion, we have added the summary section in the manuscript, see lines 5-18 in page 18 and lines 1-3 in page 19 of the change-noted manuscript.

**Some minor comments**

**Abstract:**

**1. during large storm events -> during intense geomagnetic storms**

Response:

Changed. See line 12 in page 1 of the change-noted manuscript.

**2. which cannot be FACs -> which should not be FACs**

Response:

Modified. See line 13 in page 1 of the change-noted manuscript.

**3. highly fluctuate -> high dynamic**

Response:

Revised. See line 15 in page 1 of the change-noted manuscript.

**Introduction**

**4. Page 1, line 20: "Recently, through simulations and observations, numerous studies have shown that the inner magnetosphere currents have a more complicated structure and distribution than originally thought".**

**The author are suggested to provide more detailed description of "more complicated structure and distribution than originally though", not just added references there.**

Response:

Thank you, we have provided more detailed description in the new version of manuscript. See lines 22-27 in page 1 of the change-noted manuscript.

**5. Page 1, line 26: respectively, from high latitude and low latitude -> from high and low latitudes, respectively.**

Response:

Modified. See lines 2-3 in page 2 of the change-noted manuscript.

**6. Page 2, line 6: in the latitude regions from 10 °N to 50 °N**

**In the abstract, the authors claimed that they focus on the latitude range between 10-30 °N.**

Response:

Thanks to point out our mistake, it should be from 10-30 $^\circ$ **N**. We have made a modification. See line 11 in page 2 of the change-noted manuscript.

**3. Event analysis**

**7. Page 7, line 7: (The event was once reported by Shen et al. (2014), but they only concentrated on the interval from ~07:00 to 7:25 UT). ->**

**The event was once reported by Shen et al. (2014), but they only concentrated on the interval from ~07:00 to 7:25 UT.**

Response:

Changed. See lines 7-8 in page 8 of the change-noted manuscript.

**8. Page 7, lines 11-12:**

**It can be seen that these parameters behave as same as that of the first event, but with stronger magnetic field strength.**

Response:

Modified. See lines 11-12 in page 8 of the change-noted manuscript.

**9. page 7, line 14:**

**And the largest rotation rate (Figure 3g) oscillates significantly and exhibits…**

Response:

Revised. See lines 14-15 in page 8 of the change-noted manuscript.

**4. Summary and Discussion**

**It should be Discussion and Summary**

Response:

Since we have added a summary in the last part, we modified this part as 'Discussion'.

See line 4 in page 10 of the change-noted manuscript.

**10. Page 10, line 12:**

$\varepsilon$ **and α are the particle energy and pitch angle, respectively.**

Response:

Modified. See line 12-13 in page 12 of the change-noted manuscript.

**11. Page 11, line 11:**

**During the strong storm time, turbulences, e.g., the ULF waves, result in the fluctuation of the MFLs, …**

Response:

Modified. See line 11 in page 13 of the change-noted manuscript.

**12.Provide a typical summary of this work at the end of the study.**

Response:

Provided. See lines 5-18 in page 18 and lines 1-3 in page 19 of the change-noted manuscript.

**Response to Referee #2**

We appreciate all comments from the reviewer, which help to enhance our understanding of current in this region and improve the quality of our manuscript. In this round of review, we have considered all comments seriously. The point-by-point revisions have been made and tracked in the change-noted manuscript. We hope that our revised manuscript can meet the requirement from referee and ANGEO. In the following, the comments are marked in bold Times New Roman, followed by our responses.

**The authors present an original research manuscript which aims to explain observations performed in the northern nightside magnetosphere, where northward and southward currents are detected along with oscillations in the field curvature. In this region, the dominant currents are field-aligned currents and the ring current, which should exist mostly in the X-Y-plane. The authors propose that these currents are due to curvature drifts of energetic particles. The authors base their analysis on two mid-latitude storm time vents and Cluster spacecraft data, and various analysis techniques using multiple instruments from the Cluster mission.**

**The manuscript has generally good structure, a well-chosen selection of figures and good language. The authors take into account some of the important potential sources of errors in their analysis. The topic of research is interesting, and indeed worth pursuing. However, I have some fundamental questions regarding the logic behind this analysis and the application of the methods, and would appreciate if the authors could substantially clarify the issue, as well as improve the methods which would make the results more robust.**

**Major issue: The authors present MRA-method analysis of magnetic curvature and compare it with measured current densities, postulating that the current is due to curvature drift. However, the whole method of measuring currents aboard**

**Cluster is based on the Curlometer technique, which calculates the curl (curvature) of a magnetic field and calculates the current from that. I feel that the authors need to give more reasoning as to why they consider it a new finding that one magnetic curvature analysis technique explains currents calculated via another magnetic curvature analysis technique.**

Response:

As stated in the 'Methodology' part, MRA has the ability to calculate current density, so, we use MRA method, rather than Curlometer method to directly deduce the current density in this work. So, from this point of view, we didn't use one magnetic curvature analysis technique explains currents calculated via another magnetic curvature analysis technique.

Secondly, the calculation of current density using MRA method is also based on Maxwell-Ampere's law:

$$\mathbf{J} = \frac{1}{\mu_0} \nabla \times \mathbf{B} \qquad (1)$$

From this formula, we can obtain three components of current density. However, the current density calculated from this formula includes all contributions and we cannot obtain any behind mechanism information. For example, in the inner magnetosphere, we cannot distinguish the current contribution from gradient drifts, curvature drift and the gyromotion only from this formula.

Though, Eq (1) cannot reflect any information about contributions from each part, the following three formulas can.

$$\mathbf{j}_\nabla = P_\perp \frac{\mathbf{B} \times \nabla B}{B^3},$$

$$\mathbf{j}_C = -\frac{P_\parallel}{B^2} \boldsymbol{\rho}_c \times \mathbf{B},$$

$$\mathbf{j}_G = \frac{\mathbf{B}}{B^2} \times \left[ \nabla P_\perp - \frac{P_\perp}{B} \nabla B - \frac{P_\perp}{B^2} (\mathbf{B} \cdot \nabla) \mathbf{B} \right],$$

$(2)$

So, when we combine Eq (1) and (2), it becomes possible to distinguish current density from three processes.

**Major issue: page 6 lines 13-17: If I'm interpreting this correctly, the coordinate system in use (j_B, j_N, j_R) changes constantly with the magnetic field measurements (as referred to as a local natural coordinate system in the caption of Figure 2). Is this calculated pre- or post IGRF (or dipole field?) deduction? I'm worried that the coordinate system is not well defined when the direction of the curvature changes abruptly. Looking at the evolution of the theta and phi angles for the direction of curvature, it looks to me like j_N is mostly in the east-west-direction, and has a north-south component only at the extrema of the curvature oscillation. A re-decomposition of currents into cartesian or SM coordinates would have been important to answer questions arising from this coordinate selection, and would help with evaluation of results.**

**A question that arises directly from this, is how much of the observed j_N is actually new current density in the north-south direction, and how much of it is existing ring current simply re-mapped due to an abruptly changing coordinate system?**

Response:

Thank you very much for pointing out the use of coordinate system. Yes, the coordinate system in use (j_B, j_N, j_R) changes constantly with the magnetic field measurements. And it is calculated pre-IGRF (it will be pre-dipole field in the new version of manuscript, see the following response) deduction, which intends to reflect the real background magnetic field.

To make things more clearly, we will utilize the local cylindrical coordinate system $(j_\rho, j_\varphi, j_z)$ in the new version of the manuscript, which is defined first in the work of Vallat et al. 2005 (see following Figure). $j_z$ is parallel to the $Z_{SM}$ axis; $j_\rho$ represents the radial component of the current on the plane parallel to the ($X_{SM}$, $Y_{SM}$) plane, oriented anti-earthward; $j_\varphi$ points eastward.

[Figure]

In the following Figure, we replot the three components of the current density in the new coordinate system. To reflect field aligned currents, j_B is also plotted in the figure. From $j_z$ component of the figure, it is very clear to see that the current is along southward and northward direction. Another advantage to use this coordinate system is that we can keep the consistency with Figure 5 and 6 in the manuscript (since we use z direction there).

[Figure]

The coordinate system and the corresponding description and plots have been updated in the new version of manuscript. See lines 21-23 in page 2, section 3.1 and 3.2 of the change-noted manuscript.

**Major issue: page 12, description of how the authors evaluate plasma pressure via the kappa distribution and CIS/CODIF: I believe the authors should further clarify how they perform this analysis in order to quell concerns regarding the trustworthiness of the method. I shall try to elaborate. A Kappa distribution behaves wholly differently (quasi-Maxwellian) at low energies where CODIF data is available, than at higher energies, in the tail. As seen in Figure 1, the energy spectrogram varies wildly during the storm, and thus, assuming the plasma to be in something like a steady state and describable with a kappa distribution is a bold suggestion. After all, during storms is when there is strong acceleration of particles and deviation from the mean distribution.**

**The authors could explain in better detail how is the fit performed exactly, and what are the deduced kappa values? Both parameters "a" and "kappa" are being fitted using the low-energy portion of the population - some estimation of the quality and reliability of this fit should be presented.**

Response:

Thank you. This is a critical and very interesting comment. Actually, regarding Kappa distribution, we did more works.

(1) Indeed, particles are accelerated during the storm. However, we suppose that Kappa distribution (taking the form $aE^{-\kappa}$) is still satisfied. But it should be noted that **a** and $\kappa$ are no longer a constant but varied during the storm. To verify our hypothesis, we make a test using differential flux of $H^+$ (from CIS/CODIF) for 12 Aril 2001 event. As one example, the following Figure shows the distribution function (with pitch angle is 90°) for 9 different time points during the concerned interval. It can be seen that distribution function has good scaling, which can verify our hypothesis about Kappa distribution.

[Figure]

The following Figure displays the corresponding variation of **a** and $\kappa$ during the

whole interval (using Kappa distribution). We can see that both of them are varied.

[Figure]

(2) To test if the estimated high energetic particle differential flux (using low energy particle data) is reasonable, the best way is to check with the measurement. For Cluster, RAPID can provide energy spectrograms for high energetic particles. Unfortunately, there is no RAPID data for the two concerned events. Therefore, we have tried to find another storm event to test. The selected event occurred on 20 April 2002, with the minimum of the Dst index is -149nT (we can't find a storm event with similar Dst index in the similar position). The result is presented in the following Figure. The blue dots are observations from low energetic particles (CIS/CODIF) and the red dots are measurement from high energetic particles (RAPID). The black line is the fitted result from low energetic particles. It should be noted that the observations from RAPID is limited. But still, we can see that the fitted result can basically reflect the main trend of the high energetic particles.

[Figure]

[Figure]

The above analysis demonstrates that our hypothesis about Kappa distribution is reasonable. We believe that the particle distribution during strong storm events is very complicated but interesting topic, and more detailed works are still needed, which is our next research plan. In this work, we mainly concentrate on the mechanism of the current generation. Considering the main target of this work, we think that it is better to give another detailed study and discussion for results mentioned above in the future, rather than put all figures in this manuscript. But we will add more explanation (for the above result) in the manuscript to support our calculation, see line 1 to 7 in page 16 of the change-noted manuscript.

**Also, is the population assumed to be isotropic? Equation (3) takes only the parallel pressure, after all. Figure 6 shows the final result, but comparing that with Figure 5 suggests that the pressure contribution to the calculated current is minimal, and rather, it is dominated by the curvature component (as shown in Figure 5). And if the result is dominated by curvature, then of course it will match up with the MRA method (and the inherent curlometer technique). Thus, I am not convinced that energetic particle curvature drifts are particularly important here.**

Response:

That's very interesting question. Figure 5 in the manuscript is only used to indicate

the direction variation of curvature drift current, it cannot see if curvature component dominates for the current calculation.

Actually, we have verified that the pressure contribution, rather than curvature component (as mentioned by the referee), is the dominate part for the curvature current deduction. Remember lines 11-15 in page 11 to line 1 in page 12 of the manuscript, we have mentioned that when we use low energy particle data from CIS/CODIF, the estimated curvature drift current is less than 1 nA/m$^2$. If the result dominates by curvature, it is impossible to obtain such small value. However, when we add high energy particle simulation data, the estimated current is close to current density calculated from MRA (see Figure 6 in the manuscript). It implies that the pressure is the main contribution. The role of the curvature component shown in Figure 5 in the manuscript is to keep the curvature drift current direction consistent with what we observed in Figure 2i and 3i (southward or northward current component).

**Minor issues / clarification requests:**

**page 2 line 9: Although others have called the curlometer technique "direct" measurement of the current, in truth, direct calculation would be counting charged particle fluxes. Perhaps briefly state that it uses magnetic curvature to calculate currents via Maxwell-Ampère's law.**

Response:

Thank you, as we have mentioned above, we use MRA method to calculate current density, but it also based on Maxwell-Ampère's law. We have modified the sentence. See line 15-16 in page 2 of the change-noted manuscript.

**page 2 line 20: A normal can be defined for a plane, not directly for a field line, unless you assume it to be in a plane within which the local curvature is. Please clarify.**

Response:

We make a mistake here. It should be 'binormal'. We have corrected the word. See

line 2 in page 3 of the change-noted manuscript.

**page 2 lines 24-26: Could the authors please clarify, why they state that they subtract the IGRF field, yet then proceed to describe the standard dipole formulation?**

Response:

We subtract the IGRF field because it is closer to the real magnetic field. Then, we compare with dipole field in Figure 2 and 3 because we can obtain the radius of curvature, the magnetic field gradient and rotation rates directly from Eq (1) in the manuscript. Actually, we also have tried subtracted the dipole field, the following Figure shows the calculated current density. It can be seen that the main features are the same with what we obtain from IGRF subtraction. We also have checked that the difference of the two calculated current density is less than $1nA/m^2$, so, it will not affect the conclusion of this work. To keep the consistency, we will utilize dipole field and updated the figures in the new version of manuscript. See modifications in line 7-8 in page 3 of the change-noted manuscript. Figure 2 and 3 has been updated using calculation from dipole field accordingly.

[Figure]

**page 5, Figure 2: Could the authors please explain why they plot a dipole field for comparison, instead of the IGRF field they state they use in the text?**

Response:

The reason has been explained in last response. We have unified to use only dipole field in the new version of manuscript.

**page 6 line 5: Evaluation of Figure 2 panel c shows that contrary to what is written here, the radius of curvature is nearly everywhere much greater than that of the dipole field. Only in NH3 and NH6-8 does it drop below the dipole field value, and only then in the middle of the domain.**

Response:

Thank you. The decreased radius of curvature is more visible in second case. It is more reasonable to say that the radius of curvature is varied compare with that of dipole. We have re-write the sentence in the new version of manuscript. See line 8 in page 7 of the change-noted manuscript.

**page 8 line 7: Stating that the field lines were severely deviated would be more readily confirmed had the authors included x,y,z components of the magnetic field. The radius of curvature is a challenging method of showing this, as it becomes most important at very small values, which are not clearly visible in the plots.**

Response:

From Figure 2a and Figure 3a of the manuscript, it is very easy to recognize that the magnetic field lines are severely deviated from the dipole field.

Besides, we also compared the three components of magnetic field with that of the dipole field (see following Figure). They are indeed deviated from dipole field, which is consistent with the above analysis. We didn't put the figure in the manuscript since they are too many plots.

[Figure]

[Figure]

As for radius of curvature, actually, it is a very useful method to properly show the change (stretch) of the field line. To illustrate this feature intuitively, we provide a cartoon plot in the following Figure (note that the change of real field line is exaggerated to better show the variation), for the region we concerned, the radius of curvature of the dipole field points approximately to the Earth. However, the observations of curvature direction (Figure 2d and 3d) show that the real field point to XY plane, i.e., changed (stretched) in XY plane. But, it should keep in mind that this doesn't mean that radius curvature must smaller than that of dipole field. It is completely possible larger than curvature radius of dipole field (for example, the first event).

[Figure]

**page 10 lines 16-18: The text should reference Figure 5, panels a and b. I would recommend stating more clearly what is being shown and analyzed, instead of simply referring to "a result", which here is simply the cross product of the curvature and the magnetic field. Also, the authors claim that the z component of this has the same variation trend as j_z, but j_z has not been shown in any figure. If the authors claim that this is the same as j_N, the questions regarding stability of the chosen coordinate system apply again. I think the manuscript would be**

**much improved if these doubts could be clarified.**

Response:

As has been illustrated in the above response, to keep consistency, we have utilized the new coordinate system ($j_\rho, j_\varphi, j_z$) to describe the current density this time. Now, the $j_z$ component has the same meaning in Figure 2, Figure 3, Figure 5 and Figure 6 of the manuscript.

**page 10 line 19-21: I believe the authors should clarify their reasoning for disregarding the possibility of the third term of gyromotion drift to cause currents in the j_z direction.**

Response:

According to $\mathbf{j}_G = \dfrac{\mathbf{B}}{B^2} \times \left[ \nabla P_\perp - \dfrac{P_\perp}{B} \nabla B - \dfrac{P_\perp}{B^2} (\mathbf{B} \cdot \nabla)\mathbf{B} \right]$, the gyromotion current is originated from three terms, i.e., $\mathbf{B} \times \nabla P_\perp$, $-\mathbf{B} \times \nabla B$ and $-\mathbf{B} \times (\mathbf{B} \cdot \nabla)\mathbf{B}$. Firstly, according to previous works (e.g., Lui et al., 1987; De Michelis et al., 1999), $\nabla P_\perp$ is along the radial direction (see left plot of the following Figure). For two events concerned in this work (Cluster orbit is shown in the right plot of the following Figure), $\nabla P_\perp$ should be in the X-Y plane and along the direction indicated by the red arrow. From Figure 2b and 3b in the manuscript, it is shown that the magnetic field also points to the same direction. It means that $\nabla P_\perp$ has the similar direction with magnetic field. So, the contribution from $\mathbf{B} \times \nabla P_\perp$ should be very small. Secondly, $-\mathbf{B} \times \nabla B$ is similar to the gradient drift current and can be negligible. Thirdly, since $(\mathbf{B} \cdot \nabla)\mathbf{B}$ has the same direction with $\rho_c$ ($\rho_c = (\hat{\mathbf{b}} \cdot \nabla)\hat{\mathbf{b}}, \hat{\mathbf{b}} = \mathbf{B} / |\mathbf{B}|$), according to Figure 5a and 5b, the product of $-\mathbf{B} \times (\mathbf{B} \cdot \nabla)\mathbf{B}$ (similar to $\rho_c \times \mathbf{B}$) will behave oppositely to $j_z$. Consequently, the gyromotion current has little possibility of contributing to a strong $j_z$. We have added the above explanation in line 16 of page 12 to line 9 in page

13 of the change-noted manuscript.

[Figure]

**On line 15, they stated that both the magnetic field and its gradient are pointed towards the dayside, so this term might be non-negligible.**

Response:

Since the magnetic field $\mathbf{B}$ and its gradient $\nabla B$ towards the same direction, the cross product of them ($\mathbf{B} \times \nabla B$) should approximate to zero and can be neglected.

**Figure 5: The caption states that the plot shows "results deduced from the radius of curvature of the cross magnetic field" - I would recommend the authors be more explicit and exact in their statements.**

Response:

We have re-write the captions in Figure 5. See line 3-5 in page 14 of the change-noted manuscript.

**page 13, line 6: Now the authors compare with the T96 model, but provide no reference. Does comparison with the T96 model provide some benefit over using the IGRF or dipole models, which are used(?) in the rest of the analysis? Remaining consistent would improve the readability of the manuscript.**

Response:

Thank you. We have added references for T96 model. Since the Tsyganenko model is closer to real magnetic field in the magnetosphere, it is usually used to trace footprints

of satellites (see the following Figure, which is obtained from http://ergsc.isee.nagoya-u.ac.jp/cef/orbit.cgi?jump=Submit&year=2019&dateformat=md&month=01&day=01&doy=022&period=0000&interval=1d&plottypeg=midlat&plottypem=ims&size=100). In this work, we didn't make any comparison with T96 model, but just follow the convention and use it to trace the Cluster footprints in the northern hemisphere. To avoid misunderstanding, we have reorganized the sentence in the manuscript. See line 18-20 in page 16 of the change-noted manuscript.

[Figure]

**page 13, lines 12-16: The error caused by planarity or elongation of the tetrahedron could do with a clear statement that deformation remains low. If I have understood correctly, neither the standard curlometer technique nor the MRA method attempt to remove the error, and this could be clarified.**

Response:

Yes, both these methods cannot remove the error caused by the tetrahedron. But for the result, we need to evaluate how big the error is, to guarantee it will not affect our analysis. We have clarified in line 4-5 in page 17 of the change-noted manuscript.

**page 14, Figure 7: The caption should be improved - what are the red vertical lines in panels a and b? Apparently the cross-lines in panels c and d indicate the region applicable for these two events, but this could be clearly stated - it looks**

**like the panels were identical at first glance.**

Response:

We have improved the caption, see line 2-4 in page 18 of the change-noted manuscript. The red vertical lines shown in panel a and b demarcate the concerned time interval for two events. The cross-lines in panels c and d is indeed very close, because for two events, Cluster is in the similar region and the tetrahedron shape is also similar. But there is also minor difference, which can be found in the lower horizon black line in panel c and d (see red box in the following Figure).

[Figure]

**Technical corrections:**

**page 10 line 4: The reference is incorrectly formatted; it should read "De Michelis et al., 1999"**

Response:

Modified. See line 4 in page 12 of the change-noted manuscript.

[revised manuscript text omitted]

---

## Author Response (AR2)

**Response to Referee #2**

**I would like to thank the authors for extensively improving the manuscript in agreement with my requests. The clarity of the manuscript has been improved greatly, and I would recommend publication with only a few minor corrections. In the future, however, I would urge the authors to prepare their manuscripts so that interim results such as the power-law fits, not included in the manuscript, would not be of such importance to the assessment of validity of the manuscript.**
Response:
Thank you for your suggestion. Storm time particle distribution is indeed very interesting (but complicated) topic. We will try to conduct more detailed work in the following study.

**The magnitude analysis is convincing, and the restructuring has improved the flow and focus of the manuscript significantly. The explanations from the authors have also cleared some misinterpretations on my behalf.**
Response:
Thank you. The manuscript indeed looks greatly improved after revision according to referee's suggestions.

**I would suggest the authors elaborate the abstract somewhat so it better describes the actual analysis of energetic particle fluxes.**
Response:
Elaborated, see lines 15-18 in page 1 of the change-noted manuscript.

**A few notes:**

**The fits of a and kappa to CODIF data were performed in the perpendicular direction (pitch angle of 90 degrees), whereas the equation for j_c uses parallel pressure. What was the reasoning for this? Assuming fits at pitch-angles close to 0 or 180 degrees provide similar results, I am content with the analysis, and leave this assessment to the authors.**
Response:
Actually, we also conduct the analysis for other pitch angles. They have similar results with that of 90 degrees. In the last round of response, as one example, we just showed the result for pitch angle of 90 degrees.

**For the aforementioned fits, the four dots available in RAPID data actually indicate a very different power law than what CODIF sees (if you fit to only the RAPID portion of data), although they can be placed at the continuation point of the CODIF power law. I would urge the authors to be careful of extrapolations up to 1 MeV (or beyond) based on these fits.**
Response:
It's a pity that we have only four data points on RAPID. We will make a further analysis for this issue in the future using other observations.

**minor corrections:**

**page 4, line 13: I believe this should read polar, not azimuthal? For this event, the green curves are the ones showing a difference. For the March 31st event, there is a small storm-time deviation in the azimuthal component, but there as well it's the polar angle which deviates the most, I believe.**

5  Response:

Yes, thank you for pointing out this mistake. We have made a correction ,see line 16 in page 4 of the change-noted manuscript.

**page 13, lines 5-6: I would recommend the authors add a mention that during the storm time,**
10  **currents calculated via energetic particle fluxes appear to still underestimate the current. In fact, this could be discussed in the conclusions as follows: As the particle flux fit method of calculating currents works so well earlier in the time period, this undershoot during storm time might be indicative of additional energetic particle acceleration (a harder power law) in the parallel direction. This increased parallel pressure would result in the observed larger value of j_c. It is**
15  **hypothetical, but could be verified by comparing CODIF and RAPID measurements at other storm events.**

Response:

Added, see lines 7-10 in page 13 of the change-noted manuscript.

[revised manuscript text omitted]